# Microphysical Properties of Frozen Particles Inferred from Global Precipitation Measurement (GPM) Microwave Imager (GMI) Polarimetric Measurements

Jie Gong<sup>1, 2</sup>, Dong L. Wu<sup>2</sup>

<sup>1</sup>Universities Space Research Association, Columbia, MD
 <sup>2</sup>NASA Goddard Space Flight Center, Greenbelt, MD
 *Correspondence to*: Jie Gong (Jie.Gong@nasa.gov)

melting layer and stratiform rain with passive sensors.


**Abstract.** Scattering differences induced by frozen particle microphysical properties are investigated, using the vertically (V) and horizontally (H) polarized radiances from the Global Precipitation Measurement (GPM) Microwave Imager (GMI) 89 and 166 GHz channels. It is the first study on frozen particle microphysical properties on a global scale that uses the dual-frequency microwave polarimetric signals.

From the ice cloud scenes identified by the 183.3±3 GHz channel brightness temperature (TB), we find that the scattering by frozen particles is highly polarized with V-H polarimetric differences (PD) being positive throughout the tropics and the winter hemisphere mid-latitude jet regions, including PDs from the GMI 89 and 166 GHz TBs, as well as the PD at 640 GHz from the ER-2 Compact Scanning Submillimeter-wave Imaging Radiometer (CoSSIR) during the TC4 campaign. Large polarization dominantly occurs mostly near convective outflow region (i.e., anvils or stratiform precipitation), while the polarization signal is small inside deep convective cores as well as at the remote cirrus region. Neglecting the polarimetric signal would easily result in as large as 30% error in ice water path retrievals. There is a universal "bell-curve" in the PD - TB relationship, where the PD amplitude peaks at ~ 10 K for all three channels in the tropics and increases slightly with latitude (2-4 K). Moreover, the 166 GHz PD tends to increase in the case where a melting layer is beneath the frozen particles aloft in the atmosphere, while 89 GHz PD is less sensitive than 166 GHz to the melting layer. This property creates a unique PD feature for the identification of the


Horizontally oriented non-spherical frozen particles are thought to produce the observed PD because of different ice scattering properties in the V and H polarizations. On the other hand, turbulent mixing within deep convective cores inevitably promotes the random orientation of these particles, a mechanism works effectively on reducing the PD. The current GMI polarimetric measurements themselves cannot fully disentangle the possible mechanisms.

### 1. Introduction


Cloud processes play an instrumental role in determining the surface precipitation characteristics (Rutledge and Hobbs, 1983; Gedzelman and Arnold, 1994; Igel et al., 2013; Tao et al., 2013). In particular, cloud ice processes are arguably among the most poorly understood, in part due to various ice crystal types and sizes, as well as multiple pathways in ice particle formation and evolution. While ice microphysical processes themselves cannot be readily observed from space, the integrated effects of these processes (e.g., cloud and precipitation structures, microphysical/macrophysical properties) can be inferred using remote sensing techniques (e.g., passive, active, multiple instruments, etc.).


It is imperative to distinguish between ice cloud and frozen precipitation in the atmosphere although they are both composed of various non-spherical particles (Wallace and Hobbs, 2006). Weather and climate models treat these hydrometeors quite differently by suspending ice clouds for a long duration but removing frozen precipitation unrealistically fast (Li et al., 2013). The models account ice clouds in radiation calculations but often ignore the radiative fluxes and heating rates from frozen precipitation in the air (Waliser et al., 2011). Hence, remote sensing of microphysical properties in frozen precipitation and their connection to ice clouds above and surface precipitation, like this work, will provide a valuable surrogate on climate models in representing cloud-precipitation processes.

- Retrieving microphysical properties of frozen precipitation has been a challenge for spaceborne remote sensing. It depends not only on macrophysical variables (e.g., column integrated mass amount, particle size distribution, etc.), but also on frozen particle shape and orientation (Comstock et al., 2007). Small pristine ice particles are observed with no preferred orientation when the ambient temperature is very cold, but become more horizontally oriented in warmer ice clouds (Noel and Chepfer, 2010). On the other hand, for large ice particles such as snow aggregates, it is often difficult to determine their orientation due to irregular shapes that are subject more to aerodynamic conditions than temperature (Xie, 2012).
- Passive and active microwave techniques at high-frequencies (> 89 GHz) are proven to be quite valuable for ice cloud and snowfall remote sensing because of dominance of ice scattering, whereas low frequency microwave channels are more suitable for detecting liquid precipitation and water clouds (Skofronick-Jackson and Wang, 2000; Wu and Jiang, 2004). To infer the particle shape and orientation properties, polarimetric measurements are needed and have been explored in a number of earlier studies. For example, Czekala [1998] pointed out the possibility of using off-nadir paired polarized channels at 200-340 GHz to measure horizontally oriented ice crystals. Recently, Miao et al. [2003], Xie and Miao [2011] and Xie et al. [2012] applied a
- radiative transfer model to ground-based measurements, and found up to -10K polarization differences between the vertically (V) and horizontally (H) polarimetric (V-pol and H-pol, hereafter) observations at 150 GHz during snowfall events. For spaceborne remote sensing, Prigent et al. [2001; 2005] analyzed the polarimetric observations at 37 and 85 GHz from SSMI/S (Special Sensor Microwave Imager Sounder) and 85 GHz from TRMM-TMI (Tropical Rainfall Measurement Mission Microwave
- 20 Imager), where they attributed significant polarimetric differences to horizontally oriented non-spherical liquid or frozen precipitating particles (Prigent et al., 2001; 2005). Defer et al. [2014] found similar amplitudes of polarimetric signal at 89 GHz and 157 GHz using the MADRAS (Microwave Analysis and Detection of Rain and Atmospheric Structures) instrument onboard the Megha-Tropiques Mission. Studies using ground-based radar also suggested similar polarimetric difference distribution (e.g., Homeyer and Kumjian, 2015). Davis et al. (2005) studied the 122 GHz polarimetric radiances at a very large oblique view angle
- and found noticeable differences between V- and H-pol measurements, supporting the global presence of horizontally-aligned ice particles in the upper troposphere. In a further study with Monte-Carlo simulations, Davis et al. (2007) showed such a polarimetric difference remains detectable at 190-664 GHz from large oblique angles. In addition to the microwave techniques, lidar polarimetry at visible channels has also been used to infer ice particle orientation (e.g., Hu, 2007; Noel and Chepfer, 2010; Zhou et al., 2012, 2013), but limited to the top layer of ice clouds due to poor cloud penetration with lidar techniques.
  Microwave polarimetric observations at 90-200 GHz are capable of penetrating thick clouds, and yet very much under-utilized.
  - With careful consideration of surface or liquid cloud effects, these polarimetric measurements can provide useful information on the shape and orientation of frozen particles in the atmosphere.

The launch of Global Precipitation Measurement (GPM) core satellite in February 2014 enables new investigations on microphysical properties of frozen ice particles and their connection to surface precipitation. Conically polarimetric scanning at

35 high microwave frequencies (89 and 166 GHz) from GPM's Microwave Imager (GMI) enhances sensitivity to frozen precipitation particle scattering and preserves the polarimetric information from ice clouds. Combining with GMI radiances at low microwave frequencies and GPM radar reflectivity, simultaneous retrievals of frozen particle microphysical properties (e.g., bulk size and orientation parameters) can be achieved on a global basis.

In this paper we present an analysis of the global GMI's polarimetric measurements at 89 and 166 GHz to infer the 40 microphysical properties of frozen particles above precipitating systems. In the analysis we also include the airborne 640 GHz

polarimetric observations from the NASA's Compact Scanning Submillimeter-wave Imaging Radiometer (CoSSIR) instrument to extend the sensitivity to small ice crystals. Horizontally oriented non-spherical frozen particles, and the mixing induced by vertical motion are thought as the leading causes of the observed radiance polarimetric differences. Radiative transfer models (RTMs) are employed to further quantify these hypotheses.


This paper is organized to describe the data analysis technique and radiative transfer models in Section 2, followed by the GMI and CoSSIR observation results in Section 3. The working hypotheses on the observed polarized signals are discussed in the subsequent section, with conclusions and future directions in the end.

#### 2. GMI data, analysis, and radiative transfer models

#### 2.1 **GMI** polarimetric measurements

The GPM core satellite consists of a Dual-frequency Precipitation Radar (DPR) and a passive GMI instrument with 13 channels between 10 to 190 GHz, among which the 10.65, 18.7, 36.5, 89 and 166 GHz channels have V- and H-polarizations. In this study, we use only the 89 and 166 GHz polarization data because at these high frequencies the scattering contributions from ice particles and frozen precipitation become significant. The GPM core satellite flies at an altitude of 407 km in a non-Sunsynchronized orbit, covering a latitude range of 68°S to 68°N. Its slow progressing rate over local time makes it feasible to study 15 diurnal variability of cloud and precipitation. Its wider-than-TRMM latitude coverage now allows investigations of cloud/precipitation properties in the extra-tropics. GMI has a forward conical scan off-nadir at an angle of 48.5° (52.8° incidence angle at the surface). GMI contains two sets of footprint sizes with different scan swaths. For low-frequency (LF) channels below and including 89 GHz, the scan swath is 885 km, but it becomes 835 km for high-frequency (HF) channels. Such a disparity in footprint size and swath requires a post-processing and regridding the raw radiance measurements (L1B) in order to 20 intercompare the 89 and 166 GHz polarimetric measurements. Thus, we choose to use L1C-R product instead of L1B in our study. More details on comparing the 89 and 166 GHz datasets can be found in the Appendix A.



In this study we define the polarimetric radiance difference (PD) as  $\Delta TB = TB_V - TB_H$ . Fig. 1 shows a squall line event revealed in the GMI's dual-pol 89 and 166 GHz radiances and PDs on April 29, 2014. The squall line exhibits a clear centerline of deep convective cells and periphery of anvil clouds in the radiance maps (Fig. 1a and 1b), where brightness temperature (TB) is depressed strongly in both channels but their PDs are relatively small along the deep convective line for both channels. Away from this convective center line. PDs maximize in the anvil/stratiform precipitation region (on the order of 10 K) before diminish gradually in the remote clear-sky/cirrus regions (0-5 K). This variation from convective cores to anvil outflow and further to clear sky is more obviously seen in the 166 GHz than 89 GHz maps, because of the increasing contribution of ice particle scattering at higher frequency microwave channels. It is also evident in Fig. 1c, where the 89 GHz PDs are more sensitive to polarimetric signals from ocean surfaces than the 166 GHz, exhibiting a large PD contrast between land and ocean (notice that the dark-red colored ocean observations in Fig. 1c actually way exceeded the colorbar range. In order to emphasize the common feature and disparities of the PD distribution, we chose to keep the same colorbar for Fig. 1c and 1d though). Such a land-ocean

contrast is not readily seen in the TB maps.