# Peer review of "Microphysical Properties of Frozen Particles Inferred from Global Precipitation Measurement (GPM) Microwave Imager (GMI) Polarimetric Measurements"

_Atmospheric Chemistry and Physics, 2016_

## Short Comment (SC1)

**Gong & Wu: Microphysical Properties of Frozen Particles Inferred from Global Precipitation Measurement (GPM) Microwave Imager (GMI) Polarimetric Measurements**

**JM**

November 16, 2016

This is a very important paper with some very interesting findings. The demostration of polarization differences in measurements alone is enlightning. The explanation of the pattern from a simple theoretical setup is quite impressive. I have a couple of questions, comments and concerns, though.

**1 General Comments**

My primary questions and concerns are with about the description of the radiative transfer modeling (section 2.3).

Most of all, your description of RT4 seems off in several aspects. Several points you mention are not general features (or limitations) of RT4. They might be of the specific compilation and setup that you use. In its core RT4 is a scattering solver, it is in the strict sense not a radiative transfer model: it does not provide atmospheric or particle optical properties. Evans' PolRadTran package, through which RT4 is commonly retrieved (from Evans' webpage), provides further code for creating particle optical properties though. However, this is not an inclusive part of RT4 and should be distinguished from this, i strongly think.

Furthermore, you imply that RT4 does only allow for a (single?) uniform ice layer (**p7**, **l18**:). This is wrong. The user might setup RT4 with as many layers as s/he wishes. Each layer is homogeneous, but using sufficiently many, thin layers, a non-uniform cloud can easily be modeled.

Later on, in section 4.2, you also mention and apply RT3. Would be better to have that already covered in 2.3, too. In 4.2, **p15**, **l12f**: you state "*RT3*, which allows to simulate effects from randomly orientated ice crystals". You imply here that RT4 can not simulate randomly oriented particles. This is wrong. RT4 can handle azimuthally randomly oriented particles. And completely randomly oriented particles are evidently also random in azimuth, are just one special case of azimuthally randomly oriented particles. In 4.2 you also describe RT4 as "fully polarized" model. I think this is a somewhat misleading description. RT4 actually does only calculate two Stokes components. In a plane-parallel, horizonthally homogeneous atmosphere with azimuthally randomly oriented particles, the other two components are zero, though. On **p7**, **l10f**:, you state that Yang et al. (2013) scattering properties where used. According to the paper title this only provides properties up to wavelengths of 100um. Is the title misleading, or how did you prepare your scattering data?

DDA is known to be slow in calculating scattering properties compared to other methods like Mie-theory and TMatrix-method. How do you use it to "speed up" your calculations?

Your statement of scattering properties being only weakly dependent on temperature seems in contradiction with Tang et al. (2016) (where Wu is a co-author). Could you provide some more information what refractive index model you used, and how big the "*minor*" differences are?

Does your statement "Frozen particle obey a Gamma size distribution" refer to frozen particles in general (then, I'd like to see that referenced) or to RT4 (see my general concerns above) or to your setup of the RT model in this study? Please be clear on this. I'd also like to see a reference or further details for the optimization procedure.

Apart from the RT modeling, your way of using aspect ratio needs more discussion and evidence. You first define aspect ratio as ratio of the H- and V- optical property components, which i think, is fine and could be seen just as an unfortunate terminology (as aspect ratio is commonly used for describing the geometric particle properties). However, later on you directly compare your aspect ratios with geometric aspect ratios (refering to Davis et al (2005), which in contrast to your statement find 1.2 as the best fitting AR, not 1.3) without ever discussing (or proving) whether they can be seen as equivalent.

I find your simple theoretical study very enlightning and impressive. I wonder, though, why at other places in the paper (**p15**, **19ff:**, **p17**, **119ff:**) you desparately try to find further explanations for the bell-curve when the simple study already explains such behaviour, ie more complicated explanations are not necessary.

**2 Specific comments**

**p5**, **l26ff:** You discuss a distinct branch with linear PD-TB at warm TB, later you talk about "the surface branch". I assume the further one is what you mean by surface branch, but could you make that clear?

**p6**, **l5ff:** *"It is non-trivial to determine the magnitude of PD"* – why is that? Or what do you actually meanby *"magnitude of PD"*? Is PD not simply the difference of the V- and H-channel measurements?

**p6**, **l6f**: "oceanic PDs are larger at 89GHz" – what does the comparison ("larger") refer to? larger than land PDs? larger than at 166GHz?

**p6**, **l8**: For me it is not obvious from figures 1&2 that surface emissivity is frequency dependent. It is very likely, but how is that seen in the figures? Could you elaborate on that? And also be more specific how that (freq. dependency of surface emissivity) affects the analysis of PD with respect to frozen hydrometeor microphysics?

**p6**, **l10**: You seem to imply that negative PD and/or clearsky measurements are stronger affected by noise than others. Why would they? Or do I just misread this statement?

p11, l5: Please provide a reference for the TC4 campaign.

**p11, 17ff:** "in optically thick cloud of  $TB_-V = 150K$ , which are also associated with large negative PD values" – to me Fig.5 rather looks like large negative values are all over the place, maybe a general offset for some measurements. Are these large negatives from a similar measurement time or region?

**p11, 18f:** "Data qualities are considered much noisier" – are they noisier or not? in my understanding that shouldn't be up to "consideration", but is a verifiable fact. I'd find it interesting to see the 3 days separately. Also, what is the general atmospheric situation for each of them? The cloud types observed? A reference would be good.

**p11, l20f:** "The bulk volume scattering coefficients can differ between the Vand H-polarization" – only those? what about extinction and absorption?

p14, l10f: Please provide references for the pre-dominant habit statements.

**p14, l13:** "which is indicative of stronger water vapor attenuation at 640 GHz" – could you elaborate how you come to that conclusion? to me this seems fairly far-fetched considering that so many cloud microphysics and cloud optical property aspects affect PD statistics, too.

**p16**, **l25f**: Please provide references for the different degree of orientation depnding on precipitation type.

**p17**, **l31**: How do you get to the 30% error estimate? this has not been discussed in the paper, has it?

---

## Referee Comment (RC1) · Anonymous Referee #1 · 19 Oct 2016

General comment:

This paper presents polarization observations of frozen particles from a spaceborne microwave imager. The topic is quite interesting and practical since polarization in the microwave spectral (especially >150 GHz) is still not fully understood in the research community. The paper first analyzed the observations from GPM and found some features of PD-TB over land and ocean. It then introduced radiative transfer models to further discuss and explained the "bell" curve of PD-TB for ice cloud detection, and also presented the PD with/without melting layers.

[Figure]

However, the paper is not so clear to me. Instead of discussing how the observed polarization signals can be used for further study and how this can further affect the accuracy of IWP retrievals in detail (which would be rather significant for future study), it only analyzed the observation results and only mentioned that 30% error will be caused if polarization is neglected both in the abstract and in the conclusion part. How exactly can the observation of PD improve the accuracy of IWP retrievals, how will PD be used? The results mentioned here are not convincing at all.

The structure of the paper is confusing. The authors described the observational data used in the study and analyzed the data, and then suddenly jumped to RT model description. Following the observational statistics in Section 3, another simple model was built up in Section 4. Basically, the paper was organized from data analysis to model description in Section 2, back to data statistics in Section 3, and then discussed with models again in Section 4. It seems to me that this is a bit chaotic.

At the end of Section 2.3, the authors introduced the concept of AR and defined it as the ratio of V/H scattering coefficients. Then the authors mentioned in the following sections that this parameter AR is equivalent to what is mentioned in Davis et al. 2005. However, the AR in Davis at al. 2005 describes the shape of ice particles (the ratio of the long axis to short axis of spheroids), this is totally different from the AR defined in this paper (Section 2.3 and 4.1). The authors didn't discuss in detail (1) how this AR in the paper is affected by particle microphyscis at different frequencies (habit, orientation, size and so on); (2) Why AR is independent of height, considering the complex atmospheric conditions. To understand your conclusions in the following sections, the authors should also present what the simulated value of AR is for different particles shapes/orientation/other microphysics.

I do believe the particle habit is related to the V/H scattering coefficients, but this is not the only factor. The orientation of ice particle, which was mentioned several times in the paper but not discussed here at all, is another important factor, which is related to AR defined in the paper.

[Figure]

Section 4.1 introduced a simple model to explain the "bell" relation. However, as I mentioned above, I don't understand the AR values (Equation 4 should be (T1-Tj)*(tau_2h-tau_2v) ???). Why is AR in the range of 1.1-1.3? Why is Tj-T1 roughly constant? (P13, L4). To me, Tj depends on the location of ice cloud and T1 is related to the near surface temperature. Both Tj and T1 are varying with atmospheric conditions. And basically the assumption of constant Tj-T1 is not true as the authors mentioned in the paper.

The authors claimed that the PD-TB relationship is independent of channel frequency (P13, L11-15). This conclusion is based on the assumption of constant AR in the atmosphere at different frequencies. However, this assumption wasn't proved to be true in this paper. As shown Figure 2, 3, and 5, the maximum PD corresponds to different TB and depends on channel frequency, i.e., the PD-TB relationship changes. Also Figure 8 shows the dependence of PD-TB on AR at three different frequencies, which is not roughly 1.3 for all the frequencies at it showed.

P16, L1, How accurate is the BB flag? This has a significant effect on the conclusions. Section 4.3 didn't distinguish different precipitation types: whether rain or snow. When the authors can't distinguish the liquid/frozen precipitation, the results are still too rough. No BB could be snow precipitation as the authors mentioned. For snow precipitation the snow scattering is weak and 89 GHz channels can still "see" the ocean surface. Thus the PD at 89 GHz is strong. For rain with BB, the near surface is screened by BB and rain, and the BB has polarization. Thus it could result in a higher PD as observed above. The mechanism is still too complicated and not clearly interpreted here.

Some of the figures in the paper are difficult to read. I suggest the authors to revise the figures to better understand the results. Eg. Figure 1, Left Bottom panel: you'd better use the same colorbar for comparison. It seems to me that at 89 GHz PD is also up to 12-16 K and is comparable with the 166 GHZ PD values. Figure 2, the y axis range of the left and right panels are not the same and difficult to find the right value that described in the text. Figure 4, it is not easy to read it and please optimize the figure.

Figure 10, The values of the color and contours are not described either in the figure or the figure caption.

Specific comment: 1.P1, L20. "increase slightly with latitude", How slight ? 2.P1, L25. the authors claimed that in deep convective cores, PD is reduced due to turbulence mixing. It is ambiguous, are there more ice or more liquid water ? As the authors discussed in the text, attenuation by liquid water and water vapor lead to a decrease of PD. 3.P1, L34. references are missing here. 4.P1, L37. references are missing here. Please indicate which models you mentioned here. (better name one or two). 5.P2, L6. It is not appropriate to refer to Xie et al. 2015, better cite a general one. 6.P2, L9. I didn't find this reference in the bibliography (Xie, 2012) 7.P2, L15. inappropriate references (Miao et al 2003 and Xie and Miao 2011). It would be also good to mention the paper from Defer et al. 2014. SInce they also investigated polarization signals at 157 GHz (Defer, E., V. Galligani, C. Prigent, and C. Jimenez, First observations of polarized scattering over ice clouds at close-to millimeter\ frequencies (157 GHz) with MADRAS on board the Megha-Tropiques mission, DOI : 10.1002/2014JD022353, J. Geophys. Res., 2014. ) 8.P2, L23. Davis et al. 2005 did observe polarized signals, but it is not significant as it was mentioned in the paper. 9.P6, L9, didn't find the reference in the bibliography Greenwald et al., 1997 10."habit" instead of "habitat" throughout the paper 11.P16, L22,"obsolete" or "oblate"?

---

## Referee Comment (RC2) · Anonymous Referee #2 · 5 Nov 2016

This study investigates the relation between V-H polarimetric difference (PD) and brightness temperature (TB) in high-frequency microwave channels (in particular, 89 and 166 GHz). An important discovery of this study is a universal bell-curve for the correlation between PD and TB. This bell-curve feature may be potentially useful for inferring the radiative and microphysical properties of frozen particles in the atmosphere. Furthermore, a heuristic model is designed to explore the physical mechanism of the aforementioned bell-curve.

Overall, the manuscript is well organized and clearly written. The radiative transfer

simulations and data analyses are convincing. No major technical errors are found in this manuscript. However, some revisions are suggested for the authors' consideration.

Specific comments:

1. Line 6 on page 12: the term "the aspect ratio (AR) factor" defined in this manuscript is not appropriate. Normally, the "aspect ratio" is used to indicate an ratio between two geometric dimensions along two different directions. However, in this manuscript, "the aspect ratio factor" is a quantity to quantify the difference of radiative properties associated with two polarization states. Thus, this factor should be referred to as " the dichroism factor" to indicate the difference due to different polarization states. Please see the following references:

Mishchenko MI. Extinction and polarization of transmitted light by partially aligned non-spherical grains. Astrophys J 1991; 367: 561-74.

Mishchenko MI, Travis LD, Lacis AA. Scattering, Absorption, and Emission of Light by Small Particles. Cambridge, UK: Cambridge University Press; 2002.

Parker, S. P., McGraw-Hill Dictionary of Scientific and Technical Terms, (5th Edition), McGraw-Hill, Inc., New York, 1993.

Yang, P., M. Wendisch, L. Bi, G. Kattawar, M. Mishchenko, and Y. Hu, 2011: Dependence of extinction cross-section on incident polarization state and particle orientation. J. Quant. Spectrosc. Radiat. Transfer, 112, 2035-2039.

2. This study suggests "horizontally oriented nonspherical frozen particles are thought to produce the observed PD because of different ice scattering properties in the V and H polarizations." However, previous studies based on observations in the visible channels (Noel and Chepfer 2010, Zhou et 2012, 2013) suggest that the percentage of horizontally oriented ice crystals is quite small. Apparently, further investigations are necessary validate this claim.

3. Line 11 on page 1 "It is the first study on global . . .that uses. . .": would it be better

to say "It is the first study of frozen particle microphysical properties on a global scale with the use of dual-frequency ..."

4. Line 13-14 on page 1: "the scatterings of frozen particles are": would it be better to say "the scattering by frozen particles is"

5. Line 16 on page 14 (and throughout the manuscript): "particle habitat" should be "particle habit"

---

## Author Comment (AC1)

**Reply to Reviewer#1:**

We sincerely thank the reviewer for providing his/her valuable comments and suggestions. We especially appreciate your suggestion on reordering the flow of writing to make the logic go more smoothly, as well as pointing out an important reference that we overlooked. Our replies to the questions will be shown in blue below.

Instead of discussing how the observed polarization signals can be used for further study and how this can further affect the accuracy of IWP retrievals in detail (which would be rather significant for future study), it only analyzed the observation results and only mentioned that 30% error will be caused if polarization is neglected both in the abstract and in the conclusion part. How exactly can the observation of PD improve the accuracy of IWP retrievals, how will PD be used? The results mentioned here are not convincing at all.

This point is closely related to one of the lead author's previous paper on using 157 GHz brightness temperature to retrieve column-wised ice water path (Gong and Wu, 2014). For the referee's convenience, the related figure is attached here:

Figure R1: Two-dimensional probability density function to show the empirical relationship of column-wised Ice Water Path (IWP) and 157 GHz cloud-induced brightness temperature (Tcir) relationship. This empirical relationship is generated from collocated and coincident CloudSat IWP and Microwave Humidity Sounder (MHS) Tcir at near-nadir-view (scan angle between  $-5^{\circ}$  and  $5^{\circ}$ ) measurements in the tropics ([ $25^{\circ}S$ ,  $25^{\circ}N$ ]) collected during June 2006 to March 2011. The peak showing the largest possibility is shown as the black curve. This figure is adapted from Gong and Wu, 2014, Fig. 3a. Tcir is defined as measured TB minus the clear-sky radiance (Tccr), where Tccr is calculated using the Community Radiative Transfer Model (CRTM) by inputting MERRA atmospheric profile without cloud layers.

We assume that we could reach a very similar curve for GMI's 166 GHz channel. Under this assumption, we can see that for anvil clouds (i.e., medium thick), in general Tcir falls roughly between -40 to -80 K, which corresponds to the steepest drop in the slope of the black curve in Fig. R1. Meanwhile, anvil cloud also possesses the largest PD. So a 10 K PD can easily result in 33% difference in IWP retrieval if Tcir is measured as -50K for the H-pol channel (corresponding to IWP=2 kg/m2) vs. -40K for the V-pol channel (corresponding to IWP=1.5 kg/m2). This is where our "30% of uncertainty in IWP retrieval" came from.

As this point is beyond the main ideas we intend to discuss and convey to the readers, we just mention it very briefly by the end of the main content to bring up one of the many reasons that understanding PD is important. We realize now that some readers might be interested to know

more, so we add a few sentences to clarify this point in the revised manuscript now, but readers are referred to Gong and Wu (2014) for all the subtle details.

(It's now read "Last but not the least, the observed PD-TB relationship has an important implication for cloud ice retrieval. Gong and Wu (2014) used an empirical IWP - TB relationship derived from CloudSat-MHS (Microwave Humidity Sounder) measurements for the IWP retrieval, where they found this relationship is nearly linear for medium thick ice cloud (i.e., anvils). The observed PD value range in this study therefore can be translated into a 30% IWP retrieval error if polarization is neglected. This is a very rough estimate that warrants a thorough evaluation in the future.").

The structure of the paper is confusing. The authors described the observational data used in the study and analyzed the data, and then suddenly jumped to RT model description. Following the observational statistics in Section 3, another simple model was built up in Section 4. Basically, the paper was organized from data analysis to model description in Section 2, back to data statistics in Section 3, and then discussed with models again in Section 4. It seems to me that this is a bit chaotic.

We originally followed a traditional template that puts data/model/methodology in Section 2, and results in Section 3. But as the reviewer suggested, since an example has already been given and discussed in the data description parts (Section 2.1 and 2.2), the logic flow is interrupted if we continue on Section 2.3 with model description. We now move Section 2.3 to Section 4.1, and add a paragraph at the end of the new Section 4.1 to connect the context ("In the next section, we will proceed the model explanation from an extremely simplified two-layer model. By computing the layer-by-layer radiative transfer with including the AR concept, we can reproduce the bell-curve with reasonable range of PD values. Then, the more sophisticated RTMs described above will be employed for further simulating and understanding the observed PD - TB characteristics.").

At the end of Section 2.3, the authors introduced the concept of AR and defined it as the ratio of V/H scattering coefficients. Then the authors mentioned in the following sections that this parameter AR is equivalent to what is mentioned in Davis et al. 2005. However, the AR in Davis at al. 2005 describes the shape of ice particles (the ratio of the long axis to short axis of spheroids), this is totally different from the AR defined in this paper (Section 2.3 and 4.1). The authors didn't discuss in detail (1) how this AR in the paper is affected by particle microphysics at different frequencies (habit, orientation, size and so on); (2) Why AR is independent of height, considering the complex atmospheric conditions. To understand your conclusions in the following sections, the authors should also present what the simulated value of AR is for different particles shapes/orientation/other microphysics.

I do believe the particle habit is related to the V/H scattering coefficients, but this is not the only factor. The orientation of ice particle, which was mentioned several times in the paper but

not discussed here at all, is another important factor, which is related to AR defined in the paper.

We totally agree with the reviewer (and thank you for bringing this point up) that the definition of AR in this paper is not clearly tied to a direct microphysical meaning, but rather a columnwised average ratio between  $\tau_V$  and  $\tau_H$ . Therefore, every microphysical characteristic along the line of sight that impacts the optical depth would also impact the value of AR, which includes but not limited to the particle habit, orientation and size projected to the line of sight. We now add a sentence immediately after first introducing the AR concept by stating that "As one can see from the definition, AR is a function of all microphysical property that plays a role in determining the optical depth along the line-of-sight (LOS), including particle size, orientation, habit, etc." Furthermore, we also restated when we cite Davis et al. (2005) that the definition of AR is not exactly the same between ours and theirs.

Having said that, one of the most important assumptions we made throughout the highly simplified, illustration-purposed two-layer model is that every microphysical property is homogeneous within the ice cloud layer. With that assumption, AR is equivalent to the actual axial ratio of the ice particle projected to the GPM viewing plane. Of course this is not likely the case in reality, and varies a lot case by case. But as we found later in Fig. 8, the best-fit AR varies only in a narrow range of 1.2-1.4. Considering that 89, 166 and 640 GHz channels are sensitive to completely different parts of the ice cloud layer, such a narrow range of "best-fit" value of AR strongly indicates that the homogeneity assumption along the line-of-sight is actually not bad at all in a statistical view.

Please note that in the original manuscript, when we first define "AR", we explicitly explained that "we vary the AR value but keep the rest model input parameters (e.g., Dme, IWC profile, etc.) unchanged. This is equivalent to the particle AR effect in which horizontally-oriented particles tend to create a stronger scattering for the H-pol radiation than for the V-pol". We think this statement itself (i.e., our definition of AR is equivalent to the particle AR effect) is not wrong under very stringent conditions, which are clearly stated in the context.

Section 4.1 introduced a simple model to explain the "bell" relation. However, as I men- tioned above, I don't understand the AR values (Equation 4 should be (T1-Tj)\*(tau\_2h-tau\_2v) ???). Why is AR in the range of 1.1-1.3? Why is Tj-T1 roughly constant? (P13, L4). To me, Tj depends on the location of ice cloud and T1 is related to the near surface temperature. Both Tj and T1 are varying with atmospheric conditions. And basically the assumption of constant Tj-T1 is not true as the authors mentioned in the paper.

Firstly, some typos have been corrected (mixing between V & H in subscriptions). AR=1.1, 1.2, 1.3 for the simulations to generate Fig. 7 are just some examples we chose, because the main purpose of this conceptual model is to reproduce the "bell" curve through using the concept of AR, the idea of which is then applied to the more sophisticated CRM to try to mimic the observation and to identify the "best-fit" AR.

T1-Tj essentially determines the spread of the starting point at the warmest side of the bellcurve. As one can see from the GMI observations in the tropics in Fig. 3 of the manuscript, the spread is roughly +/- 10K around 280K, so it is not a bad assumption of constant T1-Tj value. We also agree with the reviewer that it is not clearly stated in the main text about the reason we assume it is constant. We now have clarified it.

The authors claimed that the PD-TB relationship is independent of channel frequency (P13, L11-15). This conclusion is based on the assumption of constant AR in the atmosphere at different frequencies. However, this assumption wasn't proved to be true in this paper. As shown Figure 2, 3, and 5, the maximum PD corresponds to different TB and depends on channel frequency, i.e., the PD-TB relationship changes.

First of all, for 89 and 166 GHz, the analysis results shown in Fig. 3 and 5 are based on two and six months of all GMI data collected in the tropics, respectively, the sample sizes of which we believe are large enough to speak themselves out on a robust statistical sense (please note that Fig. 2 is generated only from a case study in Fig. 1, so it does not have any statistical implications whatsoever). Secondly, the PD peak amplitudes remain roughly the same (~10K) across different channel frequencies. As for the TB value where PD peaks, we admit that it is around 220K for 89 GHz, as opposed to 200K for 166 and 640 GHz. However, considering that 89 GHz contain so many surface polarization signals (e.g., the highly polarized branch in the warm TB side remains for 89 GHz even on land as shown in the top-right panel of Fig. 3), the warm side of the PD-TB relationship for 89 GHz essentially starts at a positive value, just like the Fig. 7b's situation. Please note that in this situation, the TB value where PD peaks also shifts to a warmer value (Fig. 7b). Lastly, here (P13, L11-15 in the original manuscript) we stated that "PD-TB" relationship is WEAKLY DEPENDENT on the channel frequency, not "independent". Therefore, we think the original statement is accurate and proper in tone, and we decide not to change the wording.

Also Figure 8 shows the dependence of PD-TB on AR at three different frequencies, which is not roughly 1.3 for all the frequencies at it showed.

OK, now I see what you mean. When we say PD-TB relationship is WEAKLY DEPENDENT on the channel frequency, we mean that given an AR value, the peak value of PD and where it peaks on the TB axis remain roughly unchanged (so we used the phrase "WEAKLY DEPENDENT") against frequencies. We do not by any means to implicate that this relationship is independent of AR.

We also noted that, during the revising period, Defer et al. [2014, JGR] found a somewhat smaller peak value of PD (~ 8K) at 89 GHz using the MADRAS instrument onboard the Megha-Tropiques mission, while their 157 GHz PD-TB relationship is very similar to what we found in the GMI 166 GHz. While the RTM simulations conducted in that paper concluded that PD increases with channel frequency, the authors also recognized that the simulated PD is very sensitive to particle size, density, etc. that we also found in the RT4 simulation. Therefore, RTM simulations from both of our study and Defer et al. [2014] could not lead to definitive, conclusive answers. More observations at higher-frequency channels like 640 GHz (such as ICI and our ongoing instrument development project of a polarized channel pair in the IR spectrum) are very much needed globally. This discussion has been included now in Section 5, 3rd paragraph.

P16, L1, How accurate is the BB flag? This has a significant effect on the conclusions. Section 4.3 didn't distinguish different precipitation types: whether rain or snow. When the authors can't distinguish the liquid/frozen precipitation, the results are still too rough. No BB could be snow precipitation as the authors mentioned. For snow precipitation the snow scattering is weak and 89 GHz channels can still "see" the ocean surface. Thus the PD at 89 GHz is strong. For rain with BB, the near surface is screened by BB and rain, and the BB has polarization. Thus it could result in a higher PD as observed above. The mechanism is still too complicated and not clearly interpreted here.

According to the ATBD of V1.4 L2 radar product, BB flag is quite reliable for the Ku-band. We also consulted Dr. Liang Liao in GPM team who is part of the group of developing the L2 radar retrieval product.

Please refer to the ATBD file for details:

https://pps.gsfc.nasa.gov/Documents/ATBD\_DPR\_2015\_whole\_a.pdf

Thanks for your comments on the large polarized branch in Fig. 10a. We were originally puzzled of this branch because we thought that the precipitation layer, when detectible by the Ku-band, can always effectively block the ocean surface polarization signal, but apparently it's not always the case. So we explained it by the light precipitation scenes. Unfortunately, right now we cannot tell snowfall scenes apart from the rainfall scenes, so we cannot further separate them out and interpret the results more clearly, as also noted here by the reviewer. We now include your comment in the text.

Some of the figures in the paper are difficult to read. I suggest the authors to revise the figures to better understand the results. Eg. Figure 1, Left Bottom panel: you'd better use the same colorbar for comparison. It seems to me that at 89 GHz PD is also up to 12-16 K and is comparable with the 166 GHZ PD values.

We now made the color scale the same for 89 and 166 GHz PDs.

Figure 2, the y axis range of the left and right panels are not the same and difficult to find the right value that described in the text.

The x-axis and y-axis are now made identical for easier comparison.

Figure 4, it is not easy to read it and please optimize the figure. We enlarged the font size and bolded the colored lines now.

Figure 10, The values of the color and contours are not described either in the figure or the figure caption.

Values of the color/contour scale are not important, but the total areas they cover have been normalized to unity, and plotted in log-scale. This description has been added to the figure caption. Thanks.

Specific comment:

1.P1, L20. "increase slightly with latitude", How slight ?

Other than the [-70,-50] latitude band where the results may not be significant due to limited sample size of cloudy-sky cases, the increase of the peak amplitude with latitude falls in the range of 2-4K as visually estimated from Fig. 4.

2.P1, L25. the authors claimed that in deep convective cores, PD is reduced due to turbulence mixing. It is ambiguous, are there more ice or more liquid water ? As the authors discussed in the text, attenuation by liquid water and water vapor lead to a decrease of PD. That's not quite what we meant. Turbulent mixing within deep convective core inevitably promotes the random orientation of ice, liquid and mixed-phase particles, which ultimately reduces the PD to close to 0. Now the wording has been altered to: "On the other hand, turbulent mixing within deep convective cores inevitably promotes the random orientation of these particles, a mechanism works effectively on reducing the PD."

3.P1, L34. references are missing here. Examples of different measurement techniques have been added.

4.P1, L37. references are missing here. Please indicate which models you mentioned here. (better name one or two).

We prefer not to name one or two models explicitly in the main text due to complicated reason (mainly because of funding sources). What we can say at this point is that operational model developers in the United States have realized this long-standing issue quite a while ago, and have been working diligently on changing the precipitation hydrometers forecast variables instead of the current diagnostic variables. Colleagues in Europe, especially ECMWF, have realized such a function, and that is one of the major reasons why all-sky data assimilation (clear + cloudy + precipitating scenes) is ingested better by the ECMWF model. A reference is given in the revised manuscript for interested reader.

5.P2, L6. It is not appropriate to refer to Xie et al. 2015, better cite a general one. It has been replaced by Comstock et al. [2007]. Orientation's effect on the IWC retrieval uncertainties have rarely been mentioned before though.

6.P2, L9. I didn't find this reference in the bibliography (Xie, 2012) The reference has been added. Thanks.

7.P2, L15. inappropriate references (Miao et al 2003 and Xie and Miao 2011). It would be also good to mention the paper from Defer et al. 2014. Since they also investigated polarization signals at 157 GHz (Defer, E., V. Galligani, C. Prigent, and C. Jimenez, First observations of polarized scattering over ice clouds at close-to millimeter\ frequencies (157 GHz) with MADRAS on board the Megha-Tropiques mission, DOI : 10.1002/2014JD022353, J. Geophys. Res., 2014. ) Thank you very much. We completely overlooked this reference before the revision. This paper is very informative, and we've now included it in the reference list as well as in the literature review paragraph in Section 1 and we spent a bit discussion to recognize it (paragraph 3, Section 5).

We feel like Miao et al. [2003] and Xie and Miao [2011] are appropriate to cite here in the literature review paragraph, as their obs. are based from ground and look upward, and hence the PD signal they found are more likely to be attributed to the snow layer.

8.P2, L23. Davis et al. 2005 did observe polarized signals, but it is not significant as it was mentioned in the paper.

Thanks. "Significant" has been replaced by "noticeable".

9.P6, L9, didn't find the reference in the bibliography Greenwald et al., 1997 The citation has been replaced by Wu and Jiang (2002), e.g., section 6.5.7 therein.

10. "habit" instead of "habitat" throughout the paper Revised. Thanks.

11.P16, L22,"obsolete" or "oblate"? Thanks. The typo has been corrected.

---

## Author Comment (AC2)

Reply to Reviewer#2:

Overall, the manuscript is well organized and clearly written. The radiative transfer simulations and data analyses are convincing. No major technical errors are found in this manuscript. However, some revisions are suggested for the authors' consideration.
We are grateful to the reviewer's recognition of our work, and we adapted your suggestions sincerely and carefully as shown below in blue.

Specific comments:
1. Line 6 on page 12: the term "the aspect ratio (AR) factor" defined in this manuscript is not appropriate. Normally, the "aspect ratio" is used to indicate a ratio between two geometric dimensions along two different directions. However, in this manuscript, "the aspect ratio factor" is a quantity to quantify the difference of radiative properties associated with two polarization states. Thus, this factor should be referred to as " the dichroism factor" to indicate the difference due to different polarization states. Please see the following references:
Mishchenko MI. Extinction and polarization of transmitted light by partially aligned non-spherical grains. Astrophys J 1991; 367: 561-74.
Mishchenko MI, Travis LD, Lacis AA. Scattering, Absorption, and Emission of Light by Small Particles. Cambridge, UK: Cambridge University Press; 2002.
Parker, S. P., McGraw-Hill Dictionary of Scientific and Technical Terms, (5th Edition), McGraw-Hill, Inc., New York, 1993.
Yang, P., M. Wendisch, L. Bi, G. Kattawar, M. Mishchenko, and Y. Hu, 2011: Dependence of extinction cross-section on incident polarization state and particle orientation. J. Quant. Spectrosc. Radiat. Transfer, 112, 2035-2039.
Thank you very much for bringing our attention to the history of this factor, which we didn't notice before. After reading the aforementioned references, we incline to not change the term and definition of "AR" in this paper. The "dichroism factor", based on my understanding of reading mischenko's series of papers, is associated with the geo-magnetic field which was thought in those papers being responsible for the systematic alignment. In Yang et al. [2011] paper mentioned above, they also thought "this is an optical phenomenon analogous to the dichroism".
In our paper, by assuming homogeneity of other microphysical properties along the line-of-sight, our AR is equivalent to the AR definition in Davis et al. [2005] which has a physical meaning of a ratio between the major and minor axis of a non-spheroid particle projected to the line-of-sight. We have notified this point in the manuscript (last paragraph of Section 4.2).
In addition, we now recognize the reviewer's comments on the similarity of "AR" to the "dichroism factor" and included the aforementioned citations in the 2$^{nd}$ last paragraph of new Section 4.1.

2. This study suggests "horizontally oriented nonspherical frozen particles are thought to produce the observed PD because of different ice scattering properties in the V and H polarizations." However, previous studies based on observations in the visible channels (Noel and Chepfer 2010, Zhou et 2012, 2013) suggest that the percentage of horizontally oriented ice crystals is quite small. Apparently, further investigations are necessary validate this claim.

We also noticed the related CALIPSO studies as pointed out here. However, CALIPSO is only sensitive to the very top of the ice cloud layer, the conclusion of which are therefore not implacable to be contradictory to our findings here.

Right now using the observations and RTMs provided by our current manuscript, we cannot quantify how much percentage the ice crystals are horizontally aligned. So we fully agree with the reviewer that further investigations using other observations, other channels, more sophisticated models are required to make any stronger claims. As we didn't explicitly claim anywhere in this manuscript that the horizontal alignment dominates, we though our statement in the abstract is proper to keep in the current form. Also, we mentioned immediately after that sentence that turbulent mixing (i.e., the factor determines how much percent of particles tend to be randomly oriented) likely plays another critical role in the PD-TB relationship.

3. Line 11 on page 1 "It is the first study on global . . .that uses. . .": would it be better to say "It is the first study of frozen particle microphysical properties on a global scale with the use of dual-frequency . . ."

Revised. Thanks.

4. Line 13-14 on page 1: "the scatterings of frozen particles are": would it be better to say "the scattering by frozen particles is"

Revised. Thanks.

5. Line 16 on page 14 (and throughout the manuscript): "particle habitat" should be "particle habit"

Sorry for this typo. We have corrected them.

---

## Author Comment (AC3)

Reply to Jana Mentrok:

We thank Jana very much for providing the valuable comments and suggestions, especially on the accuracy of the description of the radiative transfer models and simulations. The replies are in blue for easier reading. Some of the comments/questions have been raised by the other two anonymous reviewers, so readers are referred to read the replies to the other two reviewers.

1 General Comments

My primary questions and concerns are with about the description of the radiative transfer modeling (section 2.3). Most of all, your description of RT4 seems off in several aspects. Several points you mention are not general features (or limitations) of RT4. They might be of the specific compilation and setup that you use. In its core RT4 is a scattering solver, it is in the strict sense not a radiative transfer model: it does not provide atmospheric or particle optical properties. Evans' PolRadTran package, through which RT4 is commonly retrieved (from Evans' webpage), provides further code for creating particle optical properties though. However, this is not an inclusive part of RT4 and should be distinguished from this, I strongly think.

Thank you very much for correcting my misunderstandings and providing these constructive comments. I have now revised the RTM description section (now Section 4.1, 1$^{st}$ paragraph) adaptively.

Furthermore, you imply that RT4 does only allow for a (single?) uniform ice layer (p7, l18:). This is wrong. The user might setup RT4 with as many layers as s/he wishes. Each layer is homogeneous, but using sufficiently many, thin layers, a non-uniform cloud can easily be modeled.

What we mean is that we only assume one single uniform ice cloud layer in this study for all the RT3 and RT4 simulations. The wording has been modified to clarify that: "The RT4 simulations we carried in this study assume a uniform ice cloud layer…".

Later on, in section 4.2, you also mention and apply RT3. Would be better to have that already covered in 2.3, too. In 4.2, p15, l12f: you state "RT3, which allows to simulate effects from randomly orientated ice crystals". You imply here that RT4 can not simulate randomly oriented particles. This is wrong. RT4 can handle azimuthally randomly oriented particles. And completely randomly oriented particles are evidently also random in azimuth, are just one special case of azimuthally randomly oriented particles.

I agree with you on the fact that azimuth randomness is assumed for both RT3 and RT4, so our description is not accurate and could be misleading. Now we introduce first of RT3 in the same section of RT4, in the 3$^{rd}$ paragraph of Section 4.1.

In 4.2 you also describe RT4 as "fully polarized" model. I think this is a somewhat misleading description. RT4 actually does only calculate two Stokes components. In a plane-parallel, horizonthally homogeneous atmosphere with azimuthally randomly oriented particles, the other two components are zero, though.

Yes, I agree with you. Although in Dr. Evan's PolRadTran description page, PolRadTran was introduced as a fully-polarized RTM (http://nit.colorado.edu/polrad.html), RT4 assumes azimuthally random orientation, so that only I & Q parameters need to be calculated layer by

layer, but not for U & V. Your suggestion has now been incorporated in the 4[th] paragraph of Section 4.1

On p7, l10f:, you state that Yang et al. (2013) scattering properties where used. According to the paper title this only provides properties up to wavelengths of 100um. Is the title misleading, or how did you prepare your scattering data?
You are correct. Although Yang et al. (2013) only provides calculation for the visible to the far-infrared spectrum, we still used their calculations here for the MW frequency. Dr. Evans provided us with this configuration, and our planned next step is to use ARTS for extensive study. As you know, ARTS has integrated the RT4 solver and Liu (2008)'s MW non-spherical database, so it would be a better choice. This defect has now explicitly mentioned in the main text.

DDA is known to be slow in calculating scattering properties compared to other methods like Mie-theory and TMatrix-method. How do you use it to "speed up" your calculations?
Sorry, we meant to say "the FFT method was used to speed up the DDA calculation". It has been corrected now in the main text.

Your statement of scattering properties being only weakly dependent on temperature seems in contradiction with Tang et al. (2016) (where Wu is a co-author). Could you provide some more information what refractive index model you used, and how big the "minor" differences are?
Which Tang et al. (2016) paper were you referring to? I asked Dr. Wu and he didn't remember he co-authored a paper with Tang.
As for the refractive index model, the reference is:
Ray, P., 1972, Broadband Complex Refractive Indices of Ice and Water, Appl. Opt. 11, 1836-1844

[Figure]

One comparison of assuming different ice particle temperature (240K vs. 260K) is shown above. Only where the PDs peak on the TB-axis has changed by several Ks, but the peak amplitude of PDs barely change.

Does your statement "Frozen particle obey a Gamma size distribution" refer to frozen particles in general (then, I'd like to see that referenced) or to RT4 (see my general concerns above) or to your setup of the RT model in this study? Please be clear on this. I'd also like to see a reference or further details for the optimization procedure.

It's for RT4 and CRM simulations, and in general for many ice particle parameter retrievals, e.g., for CloudSat standard product, etc.. Evans et al. (2012) is cited because he also assumed a Gamma distribution for his TC4 retrieval.

Apart from the RT modeling, your way of using aspect ratio needs more discussion and evidence. You first define aspect ratio as ratio of the H- and V- optical property components, which i think, is fine and could be seen just as an unfortunate terminology (as aspect ratio is commonly used for describing the geometric particle properties). However, later on you directly compare your aspect ratios with geometric aspect ratios (refering to Davis et al (2005), which in contrast to your statement find 1.2 as the best fitting AR, not 1.3) without ever discussing (or proving) whether they can be seen as equivalent.

This discrepancy has also been brought up by the other two reviewers. Please refer to my replies to their related questions. In the revised manuscript, we modified significantly the 2$^{nd}$ last paragraph of Section 4.1 and the last paragraph of Section 4.2 to reflect the change.

I find your simple theoretical study very enlightning and impressive. I wonder, though, why at other places in the paper (p15, l9ff:, p17, l19ff:) you desparately try to find further explanations for the bell-curve when the simple study already explains such behaviour, ie more complicated explanations are not necessary.

Thanks for your recognition of our exploration based on the highly-simplified, conceptual model. We spent extensive efforts on trying to seek better "fitting" to the observations originate from three considerations. First of all, the conceptual model has many assumptions, e.g., one ice cloud layer, homogeneous, and there is no information about size, habit whatsoever. Furthermore, we have to assume that tau is proportional to f^4 in order to find the AR value that determines the peak PD amplitude and where it occurs on the TB axis. Apparently more sophisticated RTMs and inputs (different ice water content profile, particle size, habit, background water vapor, liquid below, etc.) are necessary to make the situation closer to the real world. Secondly and most importantly, we put every effort to seek whether we can get a deeper understanding of what factors contribute to the PD-TB relationship, and whether we can even retrieve some ice microphysical parameters (e.g., aspect ratio, effective diameter) in the future given the observational constraints. Last but not the least, RTMs and simulation settings used in this paper will serve as the touchstone for us to determine whether we can trust and extend these RTMs to a broader spectrum (e.g., higher-frequency MW beyond 640 GHz and IR) in order to find the best configuration of channels to yield the most abundant and/or most accurate ice microphysical retrieval products. This last point is closely related to our recently funded project of developing a new polarization instrument in the IR spectrum. Now we include some of the aforementioned motivations to justify our purpose of conducting extensive RTM simulations in this work (see the paragraph right below Fig. 8 after we finish

discussing the conceptual model). We don't plan to include the third point as the project has not begun yet.

2 Specific comments

p5, l26ff: You discuss a distinct branch with linear PD-TB at warm TB, later you talk about "the surface branch". I assume the further one is what you mean by surface branch, but could you make that clear?

We've changed the wording to "there is a distinct branch in Fig. 2a, showing a strong linear PD-TB$_v$ relationship at the warm or clear-sky TBs that corresponds to the surface polarization signals.".

p6, l5ff: "It is non-trivial to determine the magnitude of PD" – why is that? Or what do you actually mean by "magnitude of PD"? Is PD not simply the difference of the V- and H-channel measurements?

What we mean is the absolute value of PD.

p6, l6f: "oceanic PDs are larger at 89GHz" – what does the comparison ("larger") refer to? larger than land PDs? larger than at 166GHz?

We now reworded the sentences as "For example, as we can see from Fig. 1c, PDs are larger at 89 GHz over the ocean because of higher V-pol emissivity on calm water surfaces (acting like a mirror) than on windy surfaces, whereas land surfaces generally have little polarization due to surface roughness."

p6, l8: For me it is not obvious from figures 1&2 that surface emissivity is frequency dependent. It is very likely, but how is that seen in the figures? Could you elaborate on that? And also be more specific how that (freq. dependency of surface emissivity) affects the analysis of PD with respect to frozen hydrometeor microphysics?

After careful thinking, we remove this sentence. It is not directly derivable that surface emissivity is frequency dependent from Fig. 1 & 2.

p6, l10: You seem to imply that negative PD and/or clearsky measurements are stronger affected by noise than others. Why would they? Or do I just misread this statement?

There is a sentence immediately after this line that points to the Appendix B. We used the negative PD to estimate channel noise level.

p11, l5: Please provide a reference for the TC4 campaign.

The references (Evans et al., 2005; 2012) have been cited when we first mention the TC4 campaign and CoSSIR measurements in the last paragraph of Section 2.1.

p11, l7ff: "in optically thick cloud of TB V = 150K, which are also associated with large negative PD values" – to me Fig.5 rather looks like large negative values are all over the place, maybe a general offset for some measurements. Are these large negatives from a similar measurement time or region?

Yes, there is a sentence immediately after the one you mentioned here explains the reason:

"These cases are found in the July 19^th and August 6^th flight legs but not in July 17^th flight leg. Data qualities are considered much noisier in the former two flights than the latter one, but we still keep to show the original data from all three flight legs in Fig. 5 as the peak of PD-TB relationship alters little by including the noisier data (Frank Evans, personal communication)."

p11, l8f: "Data qualities are considered much noisier" – are they noisier or not? in my understanding that shouldn't be up to "consideration", but is a verifiable fact. I'd find it interesting to see the 3 days separately. Also, what is the general atmospheric situation for each of them? The cloud types observed? A reference would be good.

We used all 3 days of data mainly because the bell curve features stand robust for all 3 jet legs, although we also have to admit that it seems that the post-calibration does not work perfectly so some shift of the PD – TB relationship toward the negative PD is discernable, and it's hard to kick out those "bad data", as the majority still remains to have "good" quality. The 3 days of 640 GHz CoSSIR measurements have been used by Dr. Evans for simultaneous retrieval of multiple cloud ice parameters (Evans et al., 2012, ACP, Fig. 7-12), so the data quality should be good enough to be included here.

p11, l20f: "The bulk volume scattering coefficients can differ between the V- and H-polarization" – only those? what about extinction and absorption?

For the high-frequency MW channels, ice scattering dominates the extinction over the absorption, so the statement is reasonable here.

p14, l10f: Please provide references for the pre-dominant habit statements.

A reference has been included:

Libbrecht, K. G. (2005). The physics of snow crystals. Reports on Progress in Physics, 68 (4), 855-895. doi:10.1088/0034-4885/68/4/R03

p14, l13: "which is indicative of stronger water vapor attenuation at 640 GHz" – could you elaborate how you come to that conclusion? to me this seems fairly far-fetched considering that so many cloud microphysics and cloud optical prop- erty aspects affect PD statistics, too.

We agree with you that this statement is more or less at large. We now added a new paragraph at the end of Section 4.2 to include all possible explanations that make sense to us and that possible to account partially to the observed PD – TB relationship at three distinct frequencies.

p16, l25f: Please provide references for the different degree of orientation depending on precipitation type.

A reference to Bob Houze's 'Cloud Dynamics' book (Section 6.2) has been added to the citation list. Thanks.

p17, l31: How do you get to the 30% error estimate? this has not been discussed in the paper, has it?

Please check my reply to Reviewer#1's first question. The 30% error estimate was never intended to be a focus of this paper, so we just briefly mentioned at the end of the manuscript to raise up attention of some serious impacts that omitting PD could cause.